# General Compression Framework for Efficient Transformer Object Tracking

## Abstract

Transformer-based trackers have established a dominant role in the field of visual object tracking. While these trackers exhibit promising performance, their deployment on resource-constrained devices remains challenging due to inefficiencies. To improve the inference efficiency and reduce the computation cost, prior approaches have aimed to either design lightweight trackers or distill knowledge from larger teacher models into more compact student trackers. However, these solutions often sacrifice accuracy for speed. Thus, we propose a general model compression framework for efficient transformer object tracking, named CompressTracker, to reduce the size of a pre-trained tracking model into a lightweight tracker with minimal performance degradation. Our approach features a novel stage division strategy that segments the transformer layers of the teacher model into distinct stages, enabling the student model to emulate each corresponding teacher stage more effectively. Additionally, we also design a unique replacement training technique that involves randomly substituting specific stages in the student model with those from the teacher model, as opposed to training the student model in isolation. Replacement training enhances the student model's ability to replicate the teacher model's behavior. To further forcing student model to emulate teacher model, we incorporate prediction guidance and stage-wise feature mimicking to provide additional supervision during the teacher model's compression process. Our framework CompressTracker is structurally agnostic, making it compatible with any transformer architecture. We conduct a series of experiment to verify the effectiveness and generalizability of CompressTracker. Our CompressTracker-4 with 4 transformer layers, which is compressed from OSTrack, retains about **96**% performance on LaSOT (**66.1**% AUC) while achieves **2.17**$\times$ speed up.

## 1 Introduction

Visual object tracking is tasked with continuously localizing a target object across video frames based on the initial bounding box in the first frame. Transformer-based trackers have achieved promising performance on well-established benchmarks, their deployment on resource-restricted device remains a significant challenge. Developing a strong tracker with high efficiency is of great significance.

To reduce the inference cost of models, previous works attempt to design lightweight trackers or transfer the knowledge from teacher models to student trackers. Despite achieving increased speed, these existing methods still exhibit notable limitations. (1) **Inferior Accuracy.** Certain works propose lightweight tracking models [6, 10, 4, 21, 26] or employ neural architecture search (NAS) to search better architecture [42]. Due to the limited number of parameters, these models often suffer from underfitting and inferior performance. (2) **Complex Training.** Some works [15] aim to enhance the accuracy of fast trackers through transferring the knowledge from a teacher tracker to a student model.

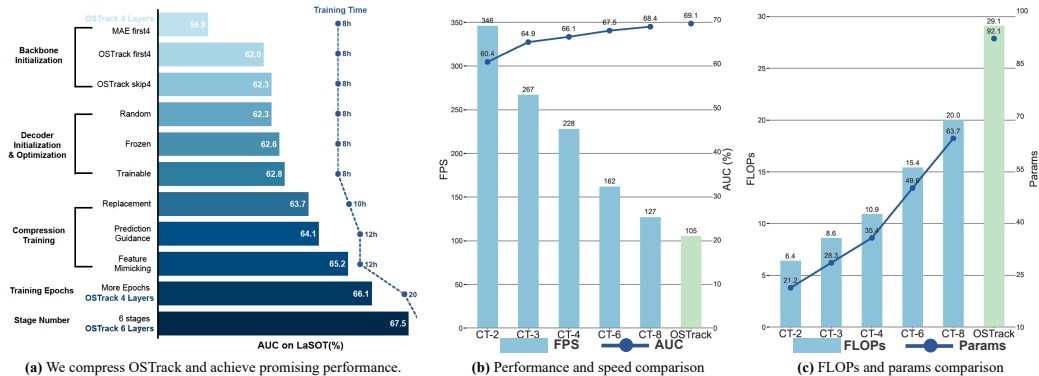

**(a)** We compress OSTrack and achieve promising performance.

**(b)** Performance and speed comparison

**(c)** FLOPs and params comparison

Figure 1: We apply our framework to OSTrack under several different layer configurations. **(a)** We implement each enhancement into our CompressTracker step by step. The training time is calculated by using 8 NVIDIA RTX 3090 GPUs. Notably, our CompressTracker-4 accelerates OSTrack by $2.17\times$ while preserving approximately $96\%$ of its original accuracy, thereby demonstrating the effectiveness of our framework. **(b)** Performance and speed comparison of CompressTracker variants with different numbers of layers. CT-x refers to a version of CompressTracker with 'x' layers. **(c)** FLOPs and parameters comparison of CompressTracker variants with different numbers of layers.

Despite the improved performance, [15] introduces a complex multi-stage training strategy, which is time-consuming. Any suboptimal performance in these individual stages can cumulatively result in suboptimal performance in the final model. (3) **Stucture Limitation.** Additionally, the model reduction paradigm in [15] severely restricts the structure of student models to be consistent only with the teacher's model.

Thus, we introduce CompressTracker, a novel and general model compression framework to enhance the efficiency of transformer tracking models. The current dominant trackers are one-stream models [44, 15, 4, 10] characterized by a series of sequential transformer encoder layers, each designed to refine the temporal matching features across frames. The output of each layer is a critical temporal matching result that is refined as the layers get deeper. Given this layer-wise refinement, it becomes a natural progression to consider the model not as a single entity but as a series of interconnected stages and encourage student tracker to align teacher model at each stage. We propose the stage division strategy, which involves partitioning the teacher model, a complex pretrained transformer-based tracking model, into distinct stages that correspond to the layers of a simpler student model. This is achieved by dividing the teacher model into a number of stages equivalent to the student model's layers. Each stage in the student model is then tasked with learning and replicating the functional behavior of its corresponding stage in the teacher model. This division is not merely a structural alteration but a strategic educational approach. By focusing each stage of the student model on mimicking a specific stage of the teacher, we enable a targeted and efficient transfer of knowledge. The student model learns not just the 'what' of tracking—i.e., the raw matching of features—but also the 'how'—i.e., the strategies developed by the teacher model at each layer of processing.

Contrary to conventional practices that isolate the training of student models, we employ a replacement training methodology that strategically intertwines the teacher and student models. The core of this methodology is the dynamic substitution of stages during training. we randomly select stages from the student model and replace them with the corresponding stages from the teacher model. By doing so, we situate the teacher model and the student model within a collaborative environment. This arrangement permits the unaltered stages of the teacher model to collaboratively inform and enhance the learning of the substituted stages in the student model rather than supervising the entire student model as a single entity. The student model is not merely learning in parallel but is directly engaging with the teacher's learned behaviors. After training, we can just combine each stage of student model for inference. The replacement training leads to a more authentic replication of the teacher's tracking strategies and helps to prevent the student model from overfitting to specific stages of the teacher model, promoting a more stable training.

To augment the learning process, we introduce prediction guidance, which serves as a supervisory signal for the student model by leveraging the teacher model's predictions. By using the predictions of the teacher model as a reference, the student model can converge more quickly. Furthermore, to enhance the similarity of the temporal matching features across corresponding stages, we have

developed a stage-wise feature mimicking strategy. This approach systematically aligns the feature representations learned at each stage of the student model with those of the teacher model, thereby promoting a more accurate and consistent learning. In Figure 1 (a), we show the procedure and the results we are able to achieve with each step toward an efficient transformer tracker.

Compared to previous works, our CompressTracker holds many merits. (1) **Enhanced Mimicking and Performance.** CompressTracker enables the student model to better mimic the teacher model, resulting in better performance. As shown in Figure 1, our CompressTracker-4 achieves $2.17\times$ speed up while maintaining about $96\%$ accuracy. (2) **Simplified Training Process.** Our CompressTracker streamlines training into a single but efficient step. This simplification not only reduces the time and resources required for training but also minimizes the potential for sub-optimal performance associated with complex procedures. The training process for CompressTracker-4 requires merely 20 hours on 8 NVIDIA RTX 3090 GPUs. (3) **Heterogeneous Model Compression.** Our stage division strategy gives a high degree of flexibility in the design of the student model. Our framework supports any transformer architecture for student model, which is not restricted to the same structure of teacher tracker. The number of layers and their structure are not predetermined but can be tailored to fit the specific computational constraints and requirements of the deployment environment.

Our contribution can be summarized as follows: (1) We introduce a novel and general model compression framework, CompressTracker, to facilitate the efficient transformer-based object tracking. (2) We propose a stage division strategy that enables a fine-grained imitation of the teacher model at the stage level, enhancing the precision and efficiency of knowledge transfer. (3) We propose the replacement training to improve the student model's capacity to replicate the teacher model's behavior. (4) We further incorporate the prediction guidance and feature mimicking to accelerate and refine the learning process of the student model. (5) Our CompressTracker breaks structural limitations, adapting to various transformer architectures for student model. It outperforms existing models, notably accelerating OSTrack [44] by $2.17\times$ while preserving approximately $96\%$ accuracy.

## 2 Related Work

**Visual Object Tracking.** Visual object tracking aims to localize the target object of each frame based on its initial appearance. Previous tracking methods [2, 28, 46, 3, 16, 27, 5, 23, 12, 41] utilize a two-stream pipeline to decouple the feature extraction and relation modeling. Recently, the one-stream pipeline hold the dominant role. [44, 14, 15, 1, 37, 8, 11, 19] combine feature extraction and relation modeling into a unified process. These models are built upon vision transformer, which consists of a series of transformer encoder layers. Thanks to a more adequate relationship modeling between template and search frame, one-stream models achieve impressive performance. However, these models suffer from low inference efficiency, which is the main obstacle to practical deployment.

**Efficient Tracking.** Some works have attempted to speed up tracking models. [42] utilizes neural architecture search (NAS) to search a light Siamese network, and the searching process is complex. [6, 10, 4, 26] design a lightweight tracking model, but the small number of parameters restricts the accuracy to a large degree. MixFormerV2 [15] propose a complex multi-stage model reduction strategy. Although MixFormerV2-S achieves real-time speed on CPU, the multi-stage training strategy is time consuming, which requires about 120 hours (5 days) on 8 Nvidia RTX8000 GPUs, even several times the original training time of MixFormer [14]. Any suboptimal performance during these stages impact the final model's performance negatively. Besides, the reduction paradigm imposes constraints on the design of student models. To address these shortcuts, we propose the general model compression framework, CompressTracker, to explore the roadmap toward an end-to-end and traininig-efficient model compression for lightweight transformer-based tracker. Our CompressTracker break the structure restriction and achieves balance between speed and accuracy.

**Transformer Compression.** Model compression aims to reduce the size and computational cost of a large model while retaining as much performance as possible, and recently many attempts have been made to speed up a large pretrained tranformer model. [18] reduced the number of parameters through pruning technique, and [35] accomplished the quantization of BERT to 2-bits utilizing Hessian information. [34, 36, 25, 38] leverage the knowledge distillation to transfer the knowledge from teacher to student model and exploit pretrained model. Beyond language models, considerable focus has also been placed on compressing vision transformer models. [33, 40, 9, 20, 7, 43, 45] utilize multiple model compression techniques to compress vision transformer models. MixFormerV2 [15]

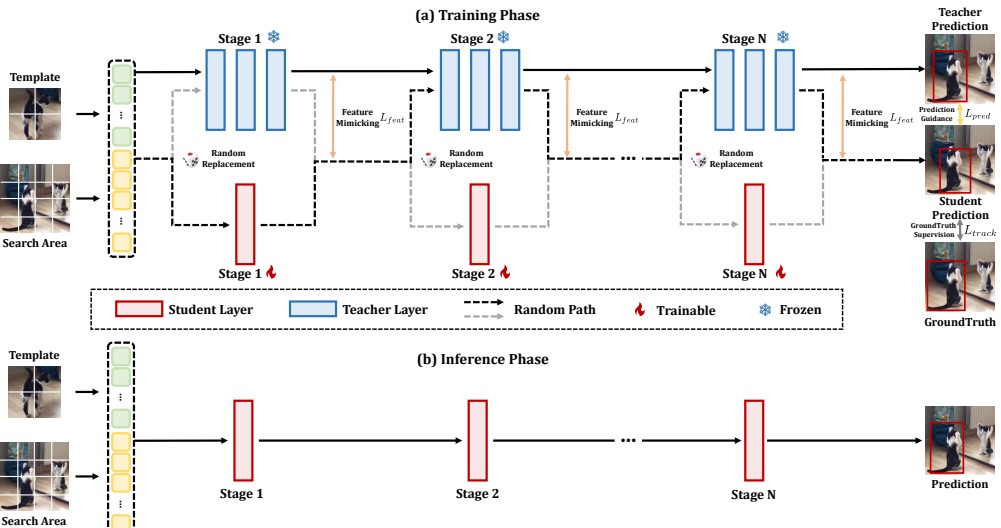

Figure 2: **CompressTracker Framework**. (a) In the training phase, we divide both the teacher model and student model into an identical number of stages. We implement a series of training strategies including replacement training, prediction guidance, and stage-wise feature mimicking, to enhance the student model's ability to emulate the teacher model. The dotted lines represent the randomly selected paths for replacement training, with black dotted lines indicating the chosen path, while grey dotted lines denote paths not selected in a specific training iteration. (b) During inference process, we simply combine each stage of the student model for testing purposes.

proposed a two-stage model reduction paradigm to distill a lightweight tracker, relying on the complex multi-stage distillation training. However, our CompressTracker propose an end-to-end and efficient compression training to achieve any transformer structure compression, which speed up OSTrack $2.17\times$ while maintaining about $96\%$ accuracy.

## 3 CompressTracker

In this section, we will introduce our proposed general model compression framework, CompressTracker. The workflow of our CompressTracker in illustrated in Figure 2.

### 3.1 Stage Division

Recently, transformer-based one-stream tracking models [8, 14, 44, 15] surpass conventional Siamese trackers [2, 13, 12], becoming the dominant manner in the field of visual object tracking. These models consist of several transformer encoder layers, each generating and progressively refining temporal matching features. Building upon this layer-wise refinement, we introduce the stage division strategy, which segments the model into a series of sequential stages. This approach encourages the student model to emulate the teacher model's behavior at each individual stage. Specifically, we denote the pretrained tracker and the compressed model as *teacher* and *student* model, with $N_t$ and $N_s$ layers, respectively. Both teacher and student models are then divided into $N_s$ stages, where each stage in the student model encompasses a single layer, and each corresponding stage in the teacher model may aggregate multiple layers. For a specific stage $i$, we establish a correspondence between the stages of the teacher and student models. The objective of stage division is to enforce each stage of the student model to replicate its counterpart in the teacher model. This stage division strategy breaks the traditional approach that treats the model as an indivisible whole [6, 10, 4, 15]. Instead, it enables a fine-grained learning process where the student model transfers knowledge from the teacher in a more detailed, stage-specific manner.

Unlike the reduction paradigm adopted in [15], which confines itself to pruning within identical structures, our CompressTracker framework facilitates support for arbitrary transformer structures of the student tracker, thanks to our innovative stage-wise division design. To align the size and channel dimensions of the student model's temporal matching features with those of the teacher model, we implement input and output projection layers before and after the student layers, respectively. These

projection layers serve as an adjustment mechanism to ensure compatibility between the teacher and student models and allow for a broader range of architectural possibilities for the student model. During the inference process, these input and output injection layers are omitted.

## 3.2 Replacement Training

During the training process, we adopt the replacement training to integrates teacher model and student models, diverging from the conventional practice of training the student model in isolation. In a specific training iteration, we implement a stochastic process to determine which stages of the student model are to be replaced by the corresponding stages of the teacher model. For the specific stage $i$, we decide whether to replace or not by random Bernoulli sampling $b_i$ with probability $p$, where $b_i \in \{0, 1\}$. If $b_i$ equals 1, the output from the preceding stage $i - 1$ is directed to the $i$ student stage, otherwise, we channel the output into the $i$ frozen teacher stage. This replacement training creates a collaborative learning environment where the teacher model dynamically supervises the student model. The unreplaced stages of teacher provide valuable contextual supervision for a specific stage in the student model. Consequently, the student model is not operating in parallel but is actively engaged with and learning from the teacher's established behaviors. For the optimization of student model, we only require the groundtruth box and denote the loss as $L_{track}$. Upon completion of the training process, the student model's stages are harmoniously combined for inference. We show the pseudocode code in Appendix A.1.

## 3.3 Prediction Guidance & Stage-wise Feature Mimicking

Replacement training enables the student model to learn the behavior of each individual stage, resulting in enhanced performance. However, merely forcing student model to emulate teacher model may be overly challenging for a smaller-sized student. Thus, we employ the teacher's predictions to further guide the learning of compressed tracker. We apply the same loss as $L_{track}$ for prediction guidance, which is denoted as $L_{pred}$. With the aid of prediction guidance, student benefits from a quicker and stable learning process, assimilating knowledge from teacher model more effectively.

While prediction guidance accelerates the convergence, the student tracker might not entirely match the complex behavior of the teacher model. We introduce the stage-wise feature mimicking to further synchronize the temporal matching features between corresponding stages of the teacher and student models. This alignment is quantified by calculating the $L_2$ distance between the outputs of these stages, which is referred as $L_{feat}$. It is worth noting that any metric assessing the discrepancy in feature distributions can serve as the loss function. However, we choose a simple $L_2$ distance rather than a complex loss to highlight the effectiveness of our stage division and replacement training strategies. The stage-wise feature mimicking not only promotes a closer similarity in the feature representations of corresponding stages but also enhances the overall coherence between the teacher and student models.

## 3.4 Progressive Replacement

In Section 3.2, we describe the replacement training strategy. Although setting the Bernoulli sampling probability $p$ as a constant value can realize the compression, these stages have not been trained together at the same time and there may be some dissonance. A further finetuning step is necessary to achieve better harmony among the stages. Thus, we introduce a progressive replacement strategy to bridges the gap between the two initially separate training phases, fostering an end-to-end easy-to-hard learning process. By adjusting the value of $p$, we can control the number of stages to be replaced. The value of $p$ gradually increases from $p_{init}$ to $1.0$, allowing for a more incremental and coherent training progression:

$$p = \begin{cases} p_{init}, & 0 <= t < \alpha_1 m, \\ p_{init} + p_{init}\frac{t - \alpha_1 m}{(1 - \alpha_1 - \alpha_2)m}, & \alpha_1 m <= t <= (1 - \alpha_2)m, \\ 1.0, & (1 - \alpha_2)m < t <= m, \end{cases} \quad (1)$$

where $m$ represents the total number of training epochs, and $t$ is a specific training epoch, $\alpha_1$ and $\alpha_2$ are hyper parameters to modulate the training process. Specifically, $\alpha_1$ controls the duration of

| Method | LaSOT | | | LaSOT$_{ext}$ | | TNL2K | | TrackingNet | | | UAV123 | | FPS |
|---|---|---|---|---|---|---|---|---|---|---|---|---|---|
| | AUC | P$_{Norm}$ | P | AUC | P | AUC | P | AUC | P$_{Norm}$ | P | AUC | P | |
| OSTrack-256 [44] | 69.1 | 78.7 | 75.2 | 47.4 | 53.3 | 54.3 | - | 83.1 | 87.8 | 82.0 | 68.3 | - | 105 |
| **CompressTracker-2** | 60.4 $_{87\%}$ | 68.5 | 61.5 | 40.4 $_{85\%}$ | 43.8 | 48.5 $_{89\%}$ | 45.0 | 78.2 $_{94\%}$ | 83.3 | 74.8 | 62.5 $_{92\%}$ | 82.5 | 346 $_{3.30\times}$ |
| **CompressTracker-3** | 64.9 $_{94\%}$ | 74.0 | 68.4 | 44.6 $_{94\%}$ | 49.6 | 52.6 $_{97\%}$ | 50.9 | 81.6 $_{98\%}$ | 86.7 | 79.4 | 65.4 $_{96\%}$ | 88.3 | 267 $_{2.54\times}$ |
| **CompressTracker-4** | 66.1 $_{96\%}$ | 75.2 | 70.6 | 45.7 $_{96\%}$ | 50.8 | 53.6 $_{99\%}$ | 52.5 | 82.1 $_{99\%}$ | 87.6 | 80.1 | 67.4 $_{99\%}$ | 88.0 | 228 $_{2.17\times}$ |
| **CompressTracker-6** | 67.5 $_{98\%}$ | 77.5 | 72.4 | 46.7 $_{99\%}$ | 52.5 | 54.7 $_{101\%}$ | 54.3 | 82.9 $_{99\%}$ | 87.8 | 81.5 | 67.9 $_{99\%}$ | 88.7 | 162 $_{1.54\times}$ |
| **CompressTracker-8** | 68.4 $_{99\%}$ | 78.0 | 73.1 | 47.2 $_{99\%}$ | 53.1 | 55.2 $_{102\%}$ | 54.8 | 83.3 $_{101\%}$ | 88.0 | 81.9 | 68.2 $_{99\%}$ | 89.0 | 127 $_{1.21\times}$ |

Table 1: **Compress OSTrack.** We compress OSTrack multiple configurations with different layer settings. CompressTracker-x denotes the compressed student model with 'x' layers. We report the performance on 5 benchmarks and calculate the performance gap in comparison to the original OSTrack. Our CompressTracker effectively achieves the balance between performance and speed.

| Method | LaSOT | | | LaSOT$_{ext}$ | | TNL2K | | TrackingNet | | | UAV123 | | FPS |
|---|---|---|---|---|---|---|---|---|---|---|---|---|---|
| | AUC | P$_{Norm}$ | P | AUC | P | AUC | P | AUC | P$_{Norm}$ | P | AUC | P | |
| MixFormerV2-B [15] | 70.6 | 80.8 | 76.2 | 50.6 | 56.9 | 57.4 | 58.4 | 83.4 | 88.1 | 81.6 | 69.9 | 92.1 | 165 |
| MixFormerV2-S [15] | 60.6 | 69.9 | 60.4 | 43.6 | 46.2 | 48.3 | 43.0 | 75.8 | 81.1 | 70.4 | 65.8 | 86.8 | 325 |
| **CompressTracker-M-S** | **62.0** $_{88\%}$ | **70.9** | **63.2** | **44.5** $_{88\%}$ | **47.1** | **50.2** $_{87\%}$ | **47.8** | **77.7** $_{93\%}$ | **82.5** | **73.0** | **66.9** $_{96\%}$ | **87.1** | 325 $_{1.97\times}$ |

Table 2: **Compress MixFormerV2.** We compress MixFormerV2 into CompressTracker-M-S with 4 layers, which is the same as MixFormerV2-S including the dimension of MLP layer. We report the performance on 5 benchmarks and calculate the performance gap in comparison to the origin MixFormerV2-B. Our CompressTracker-M-S outperforms MixFormerV2-S under the same setting.

warmup process, whereas $\alpha_2$ determines the length of final finetuning process. The mathematical expectation of $p$ for each layer is:

$$E(p) = \int_0^m p\,dt = [\frac{1 + p_{init}}{2} + \frac{1 - p_{init}}{2}(\alpha_2 - \alpha_1)]m. \tag{2}$$

It is worth noting that each layer is optimized fewer times than the total iteration count, according to the mathematical expectation. Through dynamically adjusting the replacement rate $p$, we eliminate the requirement of finetuning and accomplish an end-to-end model compression.

### 3.5 Training and Inference

Our CompressTracker is a general framework applicable to a wide array of student model architectures. For the optimization of student model, our CompressTracker solely requires an end-to-end and easy-to-hand training process instead of multi-stage training methodologies. Furthermore, our approach simplifies the loss function design, eliminating the need for complex formulations. During training, teacher model is frozen and we only optimize student tracker. The total loss for CompressTracker is:

$$L = \lambda_{track}L_{track} + \lambda_{pred}L_{pred} + \lambda_{feat}L_{feat}. \tag{3}$$

After training, the various stages of the student model are combined to create a unified model for the inference phase. Consistent with previous methods [44, 14], a Hanning window penalty is adopted.

## 4 Experiments

### 4.1 Implement Details

Our framework CompressTracker is general and not dependent on a specific transformer structure, hence we select OSTrack [44] as baseline, which is a simple and effective transformer-based tracker. The training datasets consist of LaSOT [17], TrackingNet [32], GOT-10K [24], and COCO [29], following OSTrack [44] and MixFormerV2 [15]. We set $\lambda_{track}$ as 1, $\lambda_{pred}$ as 1, and $\lambda_{feat}$ as 0.2. The $p_{init}$ is set as 0.5. We train the CompressTracker with AdamW optimizer [31], with the weight decay as $10^{-4}$ and the initial learning rate of $4 \times 10^{-5}$. The batch size is 128. The total training epochs is 500 with 60K image pairs per epoch andthe learning rate is reduced by a factor of 10 after 400 epochs. $\alpha_1$ and $\alpha_2$ are set as 0.1. The search and template images are resized to resolutions of $288 \times 288$ and $128 \times 128$. We initialize the CompressTracker with the pretrained parameters of OSTrack. We report the inference speed on a NVIDIA RTX 2080Ti GPU.

| Method | LaSOT | | | LaSOT$_{ext}$ | | TNL2K | | TrackingNet | | | UAV123 | | FPS |
|---|---|---|---|---|---|---|---|---|---|---|---|---|---|
| | AUC | P$_{Norm}$ | P | AUC | P | AUC | P | AUC | P$_{Norm}$ | P | AUC | P | |
| OSTrack-256 [44] | 69.1 | 78.7 | 75.2 | 47.4 | 53.3 | 54.3 | - | 83.1 | 87.8 | 82.0 | 68.3 | - | 105 |
| SMAT [21] | 61.7 | 71.1 | 64.6 | - | - | - | - | 78.6 | 84.2 | 75.6 | 64.3 | 83.9 | **158** |
| **CompressTracker-SMAT** | **62.8** $_{91\%}$ | **72.2** | **64.0** | **43.4** $_{92\%}$ | **46.0** | **49.6** $_{91\%}$ | **46.9** | **79.7** $_{96\%}$ | **85.0** | **75.4** | **65.9** $_{96\%}$ | **86.4** | 138 $_{1.31\times}$ |

Table 3: **Compress OSTrack for SMAT.** We compress OSTrack into CompressTracker-SAMT with 4 SMAT layers, which is the same as SMAT. We report the performance on 5 benchmarks and calculate the performance gap in comparison to the original OSTrack. Our CompressTracker-SAMT outperforms SMAT under the same setting.

| Method | LaSOT | | | LaSOT$_{ext}$ | | TNL2K | | TrackingNet | | | UAV123 | | FPS |
|---|---|---|---|---|---|---|---|---|---|---|---|---|---|
| | AUC | P$_{Norm}$ | P | AUC | P | AUC | P | AUC | P$_{Norm}$ | P | AUC | P | |
| **CompressTracker-2** | 60.4 | 68.5 | 61.5 | 40.4 | 43.8 | 48.5 | 45.0 | 78.2 | 83.3 | 74.8 | 62.5 | 82.5 | **346** |
| **CompressTracker-3** | 64.9 | 74.0 | 68.4 | 44.6 | 49.6 | 52.6 | 50.9 | 81.6 | 86.7 | 79.4 | 65.4 | 88.3 | 267 |
| **CompressTracker-4** | 66.1 | 75.2 | 70.6 | 45.7 | 50.8 | 53.6 | 52.5 | 82.1 | 87.6 | 80.1 | 67.4 | 88.0 | 228 |
| **CompressTracker-6** | 67.5 | 77.5 | 72.4 | 46.7 | 52.5 | 54.7 | 54.3 | 82.9 | 87.8 | 81.5 | 67.9 | 88.7 | 162 |
| **CompressTracker-8** | **68.4** | **78.0** | **73.1** | **47.2** | **53.1** | **55.2** | **54.8** | **83.3** | **88.0** | **81.9** | **68.2** | **89.0** | 127 |
| HiT-Base [26] | 64.6 | 73.3 | 68.1 | 44.1 | - | - | - | 80.0 | 84.4 | 77.3 | 65.6 | - | 175 |
| HiT-Samll [26] | 60.5 | 68.3 | 61.5 | 40.4 | - | - | - | 77.7 | 81.9 | 73.1 | 63.3 | - | 192 |
| HiT-Tiny [26] | 54.8 | 60.5 | 52.9 | 35.8 | - | - | - | 74.6 | 78.1 | 68.8 | 53.2 | - | 204 |
| SMAT [21] | 61.7 | 71.1 | 64.6 | - | - | - | - | 78.6 | 84.2 | 75.6 | 64.3 | 83.9 | 158 |
| MixFormerV2-S [15] | 60.6 | 69.9 | 60.4 | 43.6 | 46.2 | 48.3 | 43.0 | 75.8 | 81.1 | 70.4 | 65.8 | 86.8 | 325 |
| FEAR-L [6] | 57.9 | 68.6 | 60.9 | - | - | - | - | - | - | - | - | - | - |
| FEAR-XS [6] | 53.5 | 64.1 | 54.5 | - | - | - | - | - | - | - | - | - | 80 |
| HCAT [10] | 59.0 | 68.3 | 60.5 | - | - | - | - | 76.6 | 82.6 | 72.9 | 63.6 | - | 195 |
| E.T.Track [4] | 59.1 | - | - | - | - | - | - | 74.5 | 80.3 | 70.6 | 62.3 | - | 150 |
| LightTrack-LargeA [42] | 55.5 | - | 56.1 | - | - | - | - | 73.6 | 78.8 | 70.0 | - | - | - |
| LightTrack-Mobile [42] | 53.8 | - | 53.7 | - | - | - | - | 72.5 | 77.9 | 69.5 | - | - | 120 |
| STARK-Lightning [41] | 58.6 | 69.0 | 57.9 | - | - | - | - | - | - | - | - | - | 200 |
| DiMP [3] | 56.9 | 65.0 | 56.7 | - | - | - | - | 74.0 | 80.1 | 68.7 | 65.4 | - | 77 |
| SiamFC++ [39] | 54.4 | 62.3 | 54.7 | - | - | - | - | 75.4 | 80.0 | 70.5 | - | - | 90 |

Table 4: **State-of-the-art comparison.** We compare our CompressTracker which is compressed from OSTrack with previous light-weight tracking models. Our CompressTracker demonstrates superior performance over previous models.

## 4.2 Compress Object Tracker

In this section, we compress the pretrained OSTrack into different layer configurations. We report the performance of our CompressTracker across these configurations in Table 1. CompressTracker-4 compress OSTrack from 12 layers into 4 layers, and maintain **96**% and **99**% performance on LaSOT and TrackingNet while achieving **2.17**× speed up. Furthermore, as shown in Figure 1, the training process of CompressTracker-4 is notably efficient, requiring only approximately 20 hours using 8 NVIDIA RTX 3090 GPUs. For CompressTracker-6 and CompressTracker-8, as we increase the number of layers, the performance gap between our compresstracker and OSTrack diminishes. It is worth noting that our CompressTracker even outperforms the origin OSTrack on some benchmarks. Specifically, CompressTracker-6 reaches 54.7% AUC on TNL2K, and CompressTracker-8 achieves 55.2% AUC on TNL2K and 83.3% AUC on TrackingNet, while the origin OSTrack only achieves 54.3% AUC on TNL2K and 83.1% AUC on TrackingNet. Our framework CompressTracker demonstrates near lossless compression with the added benefit of increased processing speed.

Moreover, to affirm the generalization ability of our approach, we conduct experiments on Mix-FormerV2 [15] and SMAT [21]. MixFormerV2-S is a fully transformer tracking model consisting of 4 transformer layers, trained via a complex multi-stages model reduction paradigm. Following MixFormerV2-S, we adopt MixFormerV2-B as teacher and compress it to a student model with 4 layers. The results are shown in Table 2. Our CompressTracker-M-S share the same structure and channel dimension of MLP layers with MixFormerV2-S and outperforms MixFormerV2-S by about 1.4% AUC on LaSOT. SMAT replace the vanilla attention in transformer layer with sepa-rated attention. We compress OSTrack into a student model CompressTracker-SMAT, aligning the number and structure of transformer layer with SAMT. We maintain the decoder of OSTrack for CompressTracker-SMAT. CompressTracker-SMAT surpasses SMAT by 1.1% AUC on LaSOT, which demonstrates that our framework is flexible and not limited by the structure of transformer layer. Results in Table 1, 2, 3 verify the generalization ability and effectiveness of our framework.

## 4.3 Comparison with State-of-the-arts

To demonstrate the effectiveness of our CompressTracker, we compare our CompressTracker with state-of-the-art efficient trackers in 5 benchmarks. As shown in Table 4, our CompressTracker

| # | Init. method | AUC |
|---|---|---|
| 1 | MAE-first4 | 59.9% |
| 2 | OSTrack-first4 | 62.0% |
| 3 | OSTrack-skip4 | 62.3% |

Table 5: **Backbone Initialization.** 'MAE-first4' denotes initializing the student model using the first 4 layers of MAE-B. 'OSTrack-skip4' represents utilizing every fourth layer of OSTrack for the student model.

| # | Init. & Opt. | AUC |
|---|---|---|
| 1 | Random & Trainable | 62.3% |
| 2 | Teacher & Frozen | 62.6% |
| 3 | Teacher & Trainable | 62.8% |

Table 6: **Decoder Initialization and Optimization.** 'Random' denotes randomly initialized decoder, and 'Teacher' means the decoder is initialized with teacher parameters. 'Frozen' represents that the decoder is frozen, and 'Trainable' denotes decoder is trainable.

| # | Layer Split | AUC |
|---|---|---|
| 1 | Even | 62.8% |
| 2 | Uneven | 62.7% |

Table 7: **Stage Division.** 'Even' denotes evenly dividing stage strategy, and 'Uneven' means that the layer number of each stage in teacher model is 2,2,6,2.

| # | Epochs | AUC |
|---|---|---|
| 1 | 300 | 65.2% |
| 2 | 500 | 66.1% |

Table 8: **Training Epochs.** '300' and '500' denote the total training epochs.

Table 9: **Ablation studies on LaSOT.** The default choice for our model is colored in `gray`.

outperforms previous efficient trackers. Both HiT [26] and SMAT [21] are solely trained on the groundtruth and reduce computation through specialized network architectures. MixFormerV2-S [15] achieves model compression via a model reduction paradigm. Our CompressTracker-4 achieves 66.1% AUC on LaSOT while maintaining 228 FPS. CompressTracker-4 outperforms HiT-Base by 1.5% AUC on LaSOT without any specialized model structure design. CompressTracker-4 achieves the balance between speed and accuracy. Meanwhile, our CompressTracker-2, with just two transformer layers, maintains the highest speed at 346 FPS and also obtains competitive performance. CompressTracker-2 surpasses HiT-Tiny by 5.6% AUC on LaSOT, and achieves about the same performance as MixFormerV2-S with only two transformer layers. As we add more transformer layers with CompressTracker-6 and CompressTracker-8, we see further improvements in performance. These outcomes demonstrate the effectiveness of our CompressTracker framework.

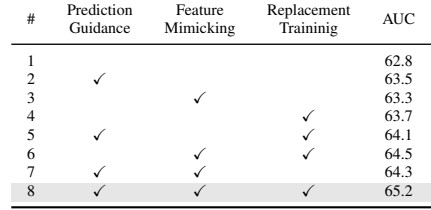

| # | Prediction Guidance | Feature Mimicking | Replacement Trainining | AUC |
|---|---|---|---|---|
| 1 | | | | 62.8 |
| 2 | ✓ | | | 63.5 |
| 3 | | ✓ | | 63.3 |
| 4 | | | ✓ | 63.7 |
| 5 | ✓ | | ✓ | 64.1 |
| 6 | | ✓ | ✓ | 64.5 |
| 7 | ✓ | ✓ | | 64.3 |
| 8 | ✓ | ✓ | ✓ | 65.2 |

Table 10: **Ablation studies** on LaSOT to analyze the supervision of student model. The default choice for our model is colored in `gray`.

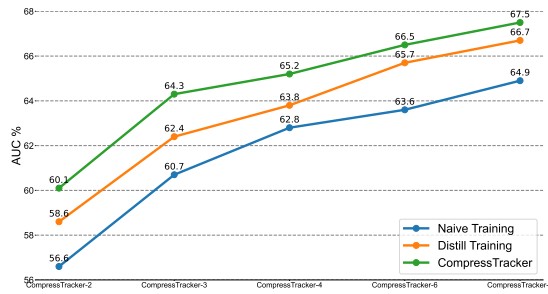

Figure 3: **Ablation study** on training strategy.

## 4.4 Ablation Study

In this section, we conduct a series of ablation studies on LaSOT to explore the factors contributing to the effectiveness of our CompressTracker. Unless otherwise specified, the teacher model is OSTrack, and the student model has 4 encoder layers. The student model is trained for 300 epochs. Please see Appendix A.2 for more analysis.

**Backbone Initialization.** We initialize the backbone of student model with different parameters and only train the student model with groundtruth supervision. The results are shown in Table 5. It can be observed that utilizing the knowledge from teacher model is crucial. Moreover, initializing with skipped layers (#3) yields slightly better performance than continuous layers. This suggests that initialization with skipped layers leads to improved representation similarity.

**Decoder Initialization and Optimization.** We investigate the influence of decoder's initialization and optimization on the accuracy of student tracker in Table 6. Initializing the decoder with parameters from the teacher model (#2) results in an improvement of approximately 0.3% compared to a decoder initialized randomly (#1), which underscores the benefits of transferring knowledge from the teacher

model to enhance the accuracy of the student model's decoder. Furthermore, making the decoder trainable leads to an additional improvement of 0.2%.

**Stage Division.** Our stage division strategy divides the teacher model into the several stages, and we explore the stage division strategy in Table 7. We design two kinds of division strategy: even and uneven, For the even division, we evenly split the teacher model's 12 layers into 4 stages, with each stage comprising 3 layers. For uneven division, we follow the design manner in [22, 30] and divide the 12 layers at a ratio of 1:1:3:1. Consequently, the number of layers in each stage of the teacher model is 2, 2, 6, and 2, respectively. The performance of the two approaches is comparable, leading us to select the equal division strategy for simplicity.

**Analysis on Supervision.** We conduct a series of experiments to comprehensively analyze the supervision effects on the student model and to verify the effectiveness of our proposed training strategy. Results are presented in Table 10. Our proposed replacement training approach (#4) improves by 0.9 % AUC compared to singly training student model on groundtruth (#1), which demonstrates that the replacement training enhances the similarity between teacher and student models. Besides, prediction guidance (#5) and feature mimicking (#8) further boost the performance, indicating the effectiveness of the two strategies. Compared to only training on groundtruth (#1), our proposed replacement training, prediction guidance and feature mimicking collectively assist student model in more closely mimicking the teacher model, resulting in a total increase of $2.4\%$ AUC.

To further explore the generalization ability of our proposed training strategy, we compare the performance of models with different layer numbers and training settings, as illustrated in Figure 3. 'Naive Training' denotes that the student model is trained without teacher supervision and replacement training. 'Distill Training' represents that the student model is trained only with teacher supervision. 'CompressTracker' refers to the same training setting in Table 10 #8. It can be observed that as the number of layers increases, there is a corresponding improvement in accuracy. Our CompressTracker shows a noticeable performance boost due to our proposed training strategy, which verifies the effectiveness and generalization ability of our framework.

**Training Epochs.** Based on the analysis in Section 3.4, the optimization steps for each layer are lower than total training steps. Thus, to ensure adequate training of each stage, we increase the training epochs from 300 to 500, and show the result in Table 8. Extending the training epochs ensures that student models receive comprehensive training, leading to improved accuracy.

# 5 Limitation

While our CompressTracker demonstrates promising performance and generalization, its training is somewhat inefficient, requiring about $2\times$ time compared to training a student model on ground truth data (20h vs. 8h on 8 NVIDIA 3090 GPUs, as shown in Figure 1 (a)). Moreover, a performance gap still exists between the teacher and student models suggests room for improvement in lossless compression. Future efforts will focus on developing more efficient training methods to boost student model accuracy and decrease training duration.

# 6 Broader Impacts

Our CompressTracker framework efficiently compresses object tracking models for edge device deployment but poses potential misuse risks, such as unauthorized surveillance. We recommend users to carefully consider the real-world implications and adopt risk mitigation strategies.

# 7 Conclusion

In this paper, we propose a general compression framework, CompressTracker, for visual object tracking. We propose a novel stage division strategy to separate the structural dependencies between the student and teacher models. We propose the replacement training to enhance student's ability to emulate the teacher model. We further introduce the prediction guidance and stage-wise feature mimicking to improve performance. Extensive experiments verify the effectiveness and generalization ability of our CompressTracker. Our CompressTracker is capable of accelerating tracking models while preserving performance to the greatest extent possible.

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

# A  Appendix / supplemental material

---

**Algorithm 1** Pseudocode of OSTrack in a PyTorch-like style

---

```
# z/x: RGB image of template/search region
# patch_embed: patch embedding layer,
# pos_embed_z/pos_embed_z: position embedding for template/search region
# blocks: transformer block layers
# decoder: decoder network

def forward(x, z):
    # patch embedding layer
    x, z = patch_embed(x), patch_embed(z)

    # add position embedding
    x, z = x + pos_embed_x, z + pos_embed_z

    # concat
    x = torch.cat([z, x], dim=1)

    # transformer layers
    for i, blk in enumerate(blocks):
        x = blk(x)

    # decode the matching result
    x = decoder(x)
```

---

## A.1  Replacement Training

We present the pseudocode for the training and testing phases of CompressTracker in Algorithm 2 and Algorithm 3, respectively. Additionally, the pseudocode of OSTrack [44] is also shown in Algorithm 1. During training process, we employ Bernoulli sampling to implement a replacement training strategy, while in the test phase, we integrate the student layers and discard the teacher layer.

| # | Replacement | AUC | Training Time |
|---|---|---|---|
| 1 | Random | 65.2% | 12 h |
| 2 | Decouple-300 | 64.6% | 16 h |

Table 11: **Ablation study** on replacement training.

| # | Replacement | AUC |
|---|---|---|
| 1 | w/ Progressive | 65.2% |
| 2 | w/o Progressive | 64.8% |

Table 12: **Ablation study** on progressive replacement.

| # | Model | Training Time |
|---|---|---|
| 1 | CompressTracker-4 | 20 h |
| 2 | OSTrack | 17 h |
| 3 | MixFormerV2-S | 120 h |

Table 13: **Training Time** comparison.

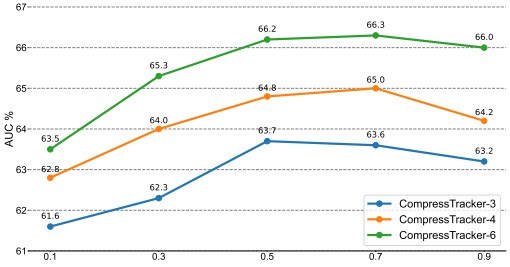

Figure 4: **Ablation study** on different replacement probability.

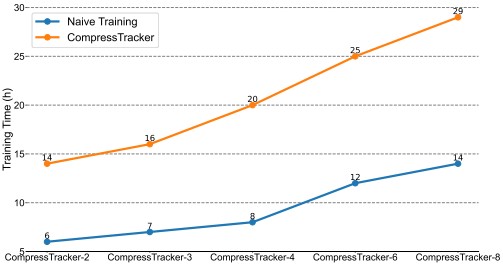

Figure 5: **Training Time.**

**Algorithm 2** Pseudocode of CompressTracker for Training in a PyTorch-like style

```python
# z/x: RGB image of template/search region
# patch_embed: patch embedding layer,
# pos_embed_z/pos_embed_z: position embedding for template/search region
# bernoulli_sample: bernoulli sampling function with probability of p
# n_s/n_t: layer number of student/teacher model
# teacher_blocks: transformer block layers of a pretrained teacher
# student_blocks: transformer block layers of student model
# decoder: decoder network

def forward(x, z):
    # patch embedding layer
    x, z = patch_embed(x), patch_embed(z)

    # add position embedding
    x, z = x + pos_embed_x, z + pos_embed_z

    # concat
    x = torch.cat([z, x], dim=1)

    # replacement sampling
    inference_blocks = []
    for i in range(n):
        if bernoulli_sample() == 1:
            inference_blocks.append(student_blocks[i])
        else:
            for j in range(n_t//n_s):
                inference_blocks.append(teacher_blocks[i*(n_t//n_s) + j])

    # randomly replaced transformer layers
    for i, blk in enumerate(inference_blocks):
        x = blk(x)

    # decode the matching result
    x = decoder(x)
```

**Algorithm 3** Pseudocode of CompressTracker for Testing in a PyTorch-like style

```python
# z/x: RGB image of template/search region
# patch_embed: patch embedding layer,
# pos_embed_z/pos_embed_z: position embedding for template/search region
# student_blocks: transformer block layers of student model
# decoder: decoder network

def forward(x, z):
    # patch embedding layer
    x, z = patch_embed(x), patch_embed(z)

    # add position embedding
    x, z = x + pos_embed_x, z + pos_embed_z

    # concat
    x = torch.cat([z, x], dim=1)

    # transformer layers
    for i, blk in enumerate(student_blocks):
        x = blk(x)

    # decode the matching result
    x = decoder(x)
```

## A.2 More Ablation Study

We represent more ablation studies on LaSOT to explore the factors contributing to effectiveness of our CompressTracker. Unless otherwise specified, teacher model is OSTrack,and student model has 4 encoder layers. The student model is trained for 300 epochs, and the $p_{init}$ is set as $0.5$.

**Replacement Training.** To evaluate the efficiency and effectiveness of our replacement training strategy, we conduct a series of experiments and results are presented in Table 11. 'Random' denotes our replacement training, and 'Decouple-300' represents decoupling the training of each stage. Result of # 1 aligns with our replacement training with 300 training epochs, while in # 2, we employ a decoupled training approach for each stage. Initially, we substitute the first stage of the teacher model with its counterpart in the student model, training the first stage for 75 epochs. Subsequently, the first trained stage of the student model is frozen, and the second stage undergoes training for an additional 75 epochs. Following this iterative process, we train the four stages cumulatively over 300 epochs, with an additional 30 epochs for fine-tuning. The 'Decouple-300' (# 2) approach achieves $64.6\%$ AUC on LaSOT with the same training epochs, marginally lower by $0.6\%$ AUC than our replacement training strategy (# 1). The 'Decouple-300' approach (# 2) requires a complex, multi-stage trainingalong with supplementary fine-tuning, while our CompressTracker operates on an end-to-end, single-step basis. Besides, the 'Decouple-300' approach may suffer from suboptimal outcomes at a specific training process, but our CompressTracker can avoid this problem through its unified training manner, which validates the superiority of our replacement training strategy.

**Replacement Probability.** We investigate the impact of replacement probability on the accuracy of student model in Figure 4. We maintain a constant replacement probability instead of implementing the progressive replacement strategy and train the student model with 300 epochs and 30 extra finetuning epochs. It can be observed from Figure 4 that performance is adversely affected when the replacement probability is set either too high or too low. Optimal results are achieved when the replacement probability is within the range of 0.5 to 0.7. Specifically, a too low probability leads to inadequate training, whereas a too high probability may result in the insufficient interaction between teacher model and student tracker. Thus, we set the $p_{init}$ as $0.5$ based on the experiment result.

**Progressive Replacement.** In Table 12, we illustrate the impact of progressive replacement strategy. The first row (# 1) corresponds to the same setting of CompressTracker, while in the second row (# 2) we fix the sampling probability as $0.5$ and the student model is trained with 300 epochs followed by 30 finetuning epochs. The absence of progressive replacement leads to a performance degradation of $0.4\%$ AUC, thereby highlighting the efficacy of our progressive replacement approach.

**Training Time.** We compare the training time of CompressTracker with 500 training epochs across different layers in Figure 5. 'Naive Training' denotes solely training on groundtruth data with 300 epochs, and 'CompressTracker' represents our proposed training strategy with 500 epochs. The training time is recorded on 8 NVIDIA RTX 3090 GPUs. Besides, the training times of our CompressTracker-4, OSTrack, and MixFormerV2-S are presented in Table 13. Although our CompressTracker requires a longer training time compared to the 'Naive Training', the increased computational overhead remains within acceptable limits. Moreover, MixFormerV2-S is trained on 8 Nvidia RTX8000 GPUs, and we estimate this will take roughly 80 hours on 8 NVIDIA RTX 3090 GPUs based on the relative computational capabilities of these GPUs. The training time of our CompressTracker-4 is significantly less than that of MixFormerV2-S, which validate the efficiency and effectiveness of our framework.

