# OpenReview forum: "General Compression Framework for Efficient Transformer Object Tracking"
_NeurIPS.cc/2024/Conference — Submitted to NeurIPS 2024_

### Official Review · Reviewer_2g8j · 2024-06-17

**Soundness:** 3
**Presentation:** 4
**Contribution:** 4
**Rating:** 9
**Confidence:** 4

**Summary:**

The authors proposed a novel compression strategy of transformer based trackers. Unlike previous works, it divides the teacher network into multiple segments, each segment corresponds a single transformer layer of student network, then train each student layer separately. It also introduced some training strategies to enhance performance including (progressive) replacement training, prediction guidance and feature mimicking. Such compression framework is insensitive to the change of architecture of teacher network.

**Strengths:**

1. Effectiveness. The experiment results clearly demonstrated significant improvement of inference speed while reserving the majority of tracking accuracy.
2. Flexibility. The proposed compression strategy is insensitive to the change of architecture of tracking models, making it easy to apply on almost any transformer based trackers. The segmentation strategy and the size of student network also supports user customization, which enables the user to design student network according to their unique demands. Such flexibility shows excellent application prospects in end-side scenarios.

**Weaknesses:**

The detailed strategy of dividing the teacher network is not stated clearly in the paper. Base on the pseudo code provided in page 13, it seems that the segmentation strategy is simply mapping the list of transformer blocks of student network to that of the teacher network base on the lengths of the two lists. This could be too simple.

For example, assume teacher network has 8 transformer blocks in module 1 and 2 blocks in module 2, while student network consists of 2 blocks, then the second student block would have to emulate the last 3 blocks of module 1 and the 2 blocks of module 2, while module 1 and  module 2 might have been trained separately and possess different knowledge. Empirically, this would result in sub-optical performance.

A brief discuss on the divide strategy could help this paper become more informative.

**Questions:**

I don't quite understand the concept of prediction guidance mentioned in the first paragraph of section 3.3: does it mean using the prediction of teacher model as pseudo label to supervise student's learning? If so, then how exactly does it help learning process, since the original ground truth bounding box already provides similar supervision, I don't really understand why using the less precise, noisy prediction of teacher tracker as additional pseudo label could actually be beneficial.

**Limitations:**

The paper clearly addressed its limitations including inefficient training process and the performance gap between teacher and student network.

---

> ### Author Rebuttal · Authors · 2024-08-06
>
> Thank you for recognizing the efficiency and value of our work. We also appreciate your insightful advice and comments. We value your support for our work!
>
>
> ## Q1: Stage Division
> Yes, our framework is indeed really simple yet effective. When the teacher model and student model share the same structure, implementing our framework requires only mapping the list of transformer layers. If student model has a different structure from teacher model, only minor modifications are needed to achieve our stage division strategy.
>
> Thanks to our stage division strategy, the number of transformer layers in each stage of teacher model is flexible. With a fixed number of stages, we can flexibly divide the layers of teacher into multiple stages. And the performance of student model relies on the layer number of student model instead of division strategy of teacher model based on our experiment experience. We have also conducted an ablation study on stage division strategy in Table 7. The 'Even' and 'UnEven' stage division strategies achieve similar performance on LaSOT.
>
> To highlight that the improvements result from our framework instead of complex stage division strategy, we just implement the even division strategy. Researchers can apply various stage division strategies based on their models. We will take your advice and add a more detailed discussion about stage division strategy in the final version.
>
>
> ## Q2: Prediction Guidance
> You are right! The prediction guidance means using the prediction of teacher model as pseudo label to supervise student's learning. While there is inherent noise in the teacher model's predictions, this noise can actually benefit the student model’s learning process. (1) Firstly, the predictions of teacher model are probability distributions, which provide additional information beyond discrete ground-truth labels. This probabilistic information helps the student model better grasp the data's underlying complexity. (2) Secondly, teacher model , having undergone extensive training, encompasses a rich set of features and patterns. By learning from the teacher model's predictions, student can learn the deep knowledge of teacher. (3) The noise in teacher prediction can enhance the generalization ability of student model. Research shows that such noisy supervision can often be more effective than using ground-truth labels alone, as it helps the student model converge more quickly and robustly [1][2][3]. In another word, although prediction of teacher contains some noise, this kind of supervision is easier to learn for the student model than groundtruth labels and can help student model converge more quickly.
>
> [1] Towards Understanding Knowledge Distillation, M. Phuong, C. Lampert, International Conference on Machine Learning, 2019
> [2] Knowledge Distillation: A Survey, J. Gou, B. Yu, S.J. Maybank, D. Tao, International Journal of Computer Vision

---

> > ### Comment · Reviewer_2g8j · 2024-08-08
> >
> > Thank you for clarifying my inquiries. Your explanations on prediction guidance have been enlightening to me, expanding my understanding of knowledge distillation, a concept I have not delved deeply into before. I appreciate your valuable contributions.

---

> > > ### Author Response · Authors · 2024-08-08
> > > **Thanks for your acknowledgement and support.**
> > >
> > > We are grateful for your recognition of the novelty and efficiency of our work. We are also very glad that our response can resolve your concerns and it's our honour! Thanks a lot for you support for our work.

---

### Official Review · Reviewer_pwR6 · 2024-07-05

**Soundness:** 3
**Presentation:** 3
**Contribution:** 2
**Rating:** 5
**Confidence:** 4

**Summary:**

This paper aims to  distill knowledge from larger teacher models into more compact student trackers. Three techniques are proposed: A stage division strategy that segments the transformer layers of the teacher model. Replacement training technique. Prediction guidance and stage-wise feature mimicking. Experiment verifys the effectiveness of the method.

**Strengths:**

1.	The proposed techniques are comprehensive and include a bunch of methods to improve the performance and efficiency of the trackers,
2.	The experiments are extensive which includes 5 VOT benchmarks.
3.	The speed is fast when applying the 2 layer tracker variants.

**Weaknesses:**

1.	The most obvious weakness is that the whole method consists of many distilling techniques, including training strategies, feature mimicking, and loss guidance. It is hard to see the inherent consistency between those techniques. This may harm the generalization ability and transferability of the proposed framework, as the author claims the framework is general.
2.	The overall method is complex. I am worried about its application to other researchers.
3.	When applied to the Mixformer v2, which has only 2 layers, performance can be improved marginally while speed is unchanged. This may indicate the method's shortcomings. Complex techniques only bring a little improvement.

**Questions:**

1.	Prove those techniques can be unified instead of looking like a bunch of tricks.
2.	The methods are restricted when the transformer tracker has fewer layers.
3.	See weakness.

**Limitations:**

Yes

---

> ### Author Rebuttal · Authors · 2024-08-06
>
> Thank you for recognizing efficiency and value of our work. We also appreciate your insightful advice and comments. We would be grateful if you could support our work and reconsider your rating.
>
>
> ## Q1: Inherent Consistency
> Thank you for your insightful advice. CompressTracker is a unified and generalized framework which can be applied to with any transformer structure. All other reviewers acknowledged the novelty, simplicity and strong generalization of our CompressTracker. we would like to clarify how our contributions are intrinsically connected and contribute to the cohesive functioning of our framework.
>
> **Motivation of stage division strategy**
>
> Current dominant trackers are one-stream models characterized by sequential transformer encoder layers that refine temporal features across frames. This layer-wise refinement suggests treating the model as interconnected stages, encouraging alignment between student and teacher at each stage. Thus, we propose **stage division strategy**, which divide teacher into distinct stages corresponding to the layers of student model. Each student stage learns and replicates the corresponding teacher stage’s function.
>
> **Motivation of replacement training**
>
> Building on this, we propose a **replacement training methodology**. The core of this method is dynamic substitution of stages during training. Thanks to our **stage division strategy**, we can perform this replacement training. Previous methods couple layers together, making replacement training impractical or potentially confusing due to strong coupling between student model stages. However, our **stage division strategy** decouples each stage, allowing replacement training and improved accuracy.
>
>
> **Motivation of prediction guidance and stage-wise feature mimicking**
>
> After that, to accelerate convergence, we introduce **prediction guidance** using the teacher's predictions as supervision. The **stage-wise feature mimicking** strategy aligns feature representations at each stage of student with those of teacher, ensuring more accurate and consistent learning.
>
>
> **Inner Connection**
>
> Our contributions are cascading and interconnected. We first propose the **stage division strategy**, which enables **replacement training**. The **replacement training** relies on our **stage division strategy**. Building on these, we introduce **prediction guidance** and **stage-wise feature mimicking strategy** to further enhance the student's learning from teacher. Each contribution lays the foundation for the next, creating a strong inherent consistency.
>
> **Generalization ability**
>
> Thanks to our **stage division strategy**, our framework has strong generalization ability, allowing flexibility in designing student model and supporting any transformer architecture. This flexibility is unique to our approach and unachievable by previous methods due to the absence of the **stage division strategy**.
>
> We conduct extensive experiments to verify effectiveness and generalization ability of our framework (Tables 1, 2, 3). In total, We have compressed **2** kinds of teacher models into **7** different student models. We experimented with **2 different teacher models**, **5 different layers of student models**, and **different structures** for student and teacher models. These student models all outperform their counterparts, demonstrating the effectiveness and generalization ability of our framework.
>
> **Simplify**
>
> Our framework is quite simple, with minimal code modifications required. We provide pseudo code in Appendix Algorithms 1, 2, and 3. Reviewers H7HB, 8Frr, and 2g8j acknowledged the simplicity, with Reviewer 2g8j expressing surprise at its simplicity. This simplicity can also prove the strong transferability of our framework.
>
> **Recognization of other Reviewers**
>
> Reviewer H7HB recognized simplicity and generalization abilities, noting, *"Versatility"* (flexbility) and *"Streamlined training"* (simplify). Reviewer M8LK highlighted novelty, stating, *"The author has clear ideas"*. Reviewer 8Frr also recognized innovation and flexbility, highlighting *"Innovative Approach"* and *"Structural Flexibility"*. Reviewer 2g8j also affirmed the flexibility.
>
> In conclusion, our contributions are inherently consistent, with each building upon the previous one. Extensive experiments verify effectiveness and generalization ability. All other reviewers acknowledged the novelty, simplicity and generalization. We will modify our manuscript to clarify the inner connection of contributions. We sincerely hope you can reconsider our work, support us, and reconsider the rating. We would appreciate it very much.
>
> ## Q2: Complex Method
> Our framework is quite simple. For details, please refer to our response to **Q1: Inherent Consistency**. We believe other researchers can reproduce our method quickly and easily. We will release all the codes upon acceptance.
>
> ## Q3: MixFormerV2-S and Fewer Transformer Layers
> Indeed, MixFormerV2-s has 4 transformer layers. We conducted an experiment to compress MixFormerV2 into CompressTracker-M-S with 4 layers, matching MixFormerV2-S. As shown in Table 2, our CompressTracker-M-S outperform MixFormerV2-S (62.0 AUC vs 60.6 AUC on LaSOT), with identical settings, including model structure, pretrained weights, and training datasets.
>
> Model performance does degrade with fewer transformer layers. CompressTracker-2 with only 2 layers remains competitive with MixFormerV2-S, which has 4 layers, and outperforms most previous models in both speed and accuracy, except for HiT-Base and SAMT, as shown in Table 4. Additionally, MixFormerV2-S requires a complex multi-stage training process, taking 120 hours, while CompressTracker-2 achieves similar or better results with just 14 hours training, as shown in Appendix Figure 5.
>
> We will add more comparision with MixFormerV2-S in the final version to further emphasize effectiveness and simplicity of our CompressTracker.

---

> > ### Comment · Reviewer_pwR6 · 2024-08-08
> >
> > Thanks for your responses. Though reducing the layer numbers of the vit-based tracking is not new, the contribution is non-trivial. I acknowledge the contribution of this work. Thus, I am willing to raise my initial rating to 5 (bordline accept).

---

> > > ### Author Response · Authors · 2024-08-08
> > > **Thanks for your acknowledgement and support.**
> > >
> > > We are grateful for your recognition of our contribution. Thanks a lot for your reconsidering the rating. We will work further to make our work better. Thanks a lot for you support for our work.

---

### Official Review · Reviewer_8Frr · 2024-07-11

**Soundness:** 3
**Presentation:** 3
**Contribution:** 2
**Rating:** 6
**Confidence:** 4

**Summary:**

The paper introduces CompressTracker, a novel general model compression framework that enhances the efficiency of transformer-based object tracking models. It innovatively segments transformer layers into stages, enabling a more effective emulation of complex teacher models by lightweight student models. The framework incorporates a unique replacement training technique, prediction guidance, and feature mimicking to refine the student model's performance. Extensive experiments demonstrate CompressTracker's effectiveness in significantly speeding up tracking models with minimal loss of accuracy, showcasing its potential for real-time applications on resource-constrained devices.

**Strengths:**

1）	Innovative Approach: The paper presents a novel compression framework, CompressTracker, which innovatively addresses the challenge of deploying transformer-based trackers on resource-limited devices by significantly reducing model size and computation cost without substantial loss of accuracy.

2）Structural Flexibility: A key advantage of the proposed framework is its structural agnosticism, allowing it to be compatible with any transformer architecture. This flexibility enables the adaptation of CompressTracker to various student model configurations, catering to diverse deployment environments and computational constraints.

3）Efficiency and Performance: The paper demonstrates through extensive experiments that CompressTracker achieves a remarkable balance between inference speed and tracking accuracy. It notably accelerates the tracking process while maintaining high performance levels, as evidenced by the nearly 96% retention of original accuracy with a 2.17× speedup.

**Weaknesses:**

1）The concept of "prediction guidance and stage-wise feature mimicking" and the idea of BEVDistill [1] seem somewhat similar.

2）Despite the model's efficiency in inference, the training process for CompressTracker is relatively inefficient.

3）While the paper shows promising results on certain benchmarks, there may be concerns about how well these findings generalize across different types of tracking tasks and real-world scenarios.

4）The paper does not compare with other model compression techniques, such as knowledge distillation, model quantization, and pruning.

5）According to the results in Table 3, I observed that the outcomes of CompressTracker-2 are inferior to those of MixFormerV2-S. What could be the reason for this?

6）It is necessary to apply compression to other tracking models in order to further validate the efficacy of the CompressTracker presented in this paper.

7）The authors lack a sufficiently comprehensive review of the related work. The authors should give more reasonable related work by carefully introducing the recent approaches to tracking with compression, such as [2].

[1] BEVDistill: Cross-Modal BEV Distillation for Multi-View 3D Object Detection, ICLR 2023.

[2] Distilled Siamese Networks for Visual Tracking, TPAMI 2021.

**Questions:**

Please refer to weaknesses.

**Limitations:**

Please refer to weaknesses.

---

> ### Author Rebuttal · Authors · 2024-08-06
>
> Thank you for your support and insights. We appreciate your deep understanding of the innovation and effectiveness of our work. We are grateful for your support for our work!
>
> ## Q1: Difference from BEVDistill
> Our CompressTracker is quite different from BEVDistill. (1) Purpose and Scope. CompressTracker is designed for visual object tracking, whereas BEVDistill transfers depth information from LiDAR to image backbones. (2) Feature Alignment. BEVDistill aligns BEV features from LiDAR and image encoders, while CompressTracker enforces that each stage in the student model’s encoder mimics the corresponding stage in the teacher model. Unlike BEVDistill, which runs encoders separately, CompressTracker performs feature mimicking at each encoder stage. (3)Distillation Approach. BEVDistill employs a complex sparse instance distillation method, whereas CompressTracker only uses a simple loss function for supervision.
>
> In a summary, our CompressTracker differs from BEVDistill in the motivation, target of distillation, implement details, and supervision method.
>
> ## Q2: Training Inefficiency
> We acknowledge the training inefficiency in Limitation (Section 5). To the best of our knowledge, our CompressTracker achieves the best balance between accuracy and inference speed. While some methods train lightweight models directly on groundtruth, resulting in less training time but lower performance, CompressTracker outperforms these models, such as HiT and SMAT, in accuracy and maintains competitive or superior inference speed (Table 4).
>
> In contrast, other methods, like MixFormerV2, use complex multi-stage training strategies, leading to much longer training times. Appendix Table 13 shows that our CompressTracker-4 (20 hours) requires only about 1/6 of the training time of MixFormerV2-S (120 hours) and achieves a 5.5 AUC improvement on LaSOT over MixFormerV2-S.
>
> Although our CompressTracker have a slightly longer training time, CompressTracker surpasses previous models and achieves the best balance between inferece speed and accuracy. We will develop more efficient training methods to enhance accuracy and decrease training duration.
>
> ## Q3: Generalization Ability
> Our CompressTracker demonstrates strong performance and generalization across various benchmarks and real-world scenarios. We conduct additional experiments on datasets like DepthTrack, which includes depth images from challenging conditions, and OTB, another popular RGB benchmark.
>
> As shown in the table, our CompressTracker-4 consistently maintains high performance across these datasets, highlighting its robustness and generalization ability. We will add experiments on more benchmarks in the final version to verify the generalizaiton ability.
>
> | Dataset | OSTrack | CompressTracker-4 |
> |-------|-------|-------|
> | DepthTrack | 51.5 | 49.7 |
> | OTB | 69.4 | 68.4 |
>
> ## Q4: Other Model Compression Techniques
> We have compared CompressTracker with several model compression techniques in our paper. As shown in Table 10 and Figure 3, our CompressTracker outperforms knowledge distillation (row #7 in Table 7 and 'Distill Training' in Figure 3) and surpasses MixFormerV2-S by 5.5 AUC on LaSOT, despite MixFormerV2-S using pruning for speedup (Table 4).
>
> We appreciate your suggestion and will add experiments comparing other model compression techniques, such as model quantization, in the final version of the manuscript.
>
> ## Q5: CompressTracker-2
> CompressTracker-2 and MixFormerV2-S exhibit different performances across various datasets. CompressTracker-2 outperform MixFormerV2-S by 2.4 AUC on TrackingNet, while MixFormerV2-S surpasses CompressTracker-2 by 3.3 AUC on UAV123 and 3.2 AUC on LaSOT extension. Both models perform similarly on LaSOT and TNL2K. It is noteworthy that CompressTracker-2 uses only 2 transformer blocks, whereas MixFormerV2-S includes 4 transformer blocks, similar to our CompressTracker-4.
>
> We also conduct experiment to compress MixFormerV2 into CompressTracker-M-S, which shares the same structure with MixFormerV2-S. As shown in Table 2. Our CompressTracker-M-S substantially outperforms MixFormerV2-S across five datasets while maintaining the same speed. Additionally, MixFormerV2-S’s multi-stage training is more time-consuming (120 hours) compared to CompressTracker’s 20 hours for CompressTracker-4, as shown in Appendix Table 13.
>
> Morevoer, the model reduction paradigm utilized by MixFormerV2-S restricts the structure of student models to be consistent only with the teacher’s model, while our CompressTracker framework supports a diverse range of any transformer architectures for student, thanks to our stage division. We will add more analysis to highlight the advantage of CompressTracker in the final version.
>
> ## Q6: Applied to Other Tracker
> Yes! Applying compression to other tracking models is crucial for further validating CompressTracker's efficacy. We have conducted extensive experiments to assess the effectiveness and generalization ability of CompressTracker. We compress OSTrack into different layers in Table 1, and compress MixFormerV2 into CompressTracker-M-S, which is the same as MixFormerV2-S, in Table 2. Besides, we also compress OSTrack into CompressTracker-SAMT with 4 SMAT layers, which is the same as SMAT, in Table 3.
>
> In total, We have compressed **2** kinds of teacher models into **7** different student models with varying model structures. Our CompressTrackers with different configurations outperform their counterparts, demonstrating effectiveness and generalization ability of CompressTracker. We will add additional experiments applying compression to more tracking models to further validate the efficacy of our CompressTracker in the final version.
>
> ## Q7: Related Work
> Thanks for your suggestion, we will add additional reviews of related work, such as [1], to better clarify the difference between CompressTracker and previous works in the final version.
>
> [1] Distilled Siamese Networks for Visual Tracking, TPAMI 2021.

---

> > ### Comment · Reviewer_8Frr · 2024-08-14
> >
> > Thanks for the authors. All concerns have been addressed. I will keep my rating.

---

> > > ### Author Response · Authors · 2024-08-14
> > >
> > > We are grateful for your recognition of the novelty and efficiency of our work. We are glad that our response has addressed your concerns. Thanks a lot for you support for our work.

---

### Official Review · Reviewer_M8LK · 2024-07-12

**Soundness:** 3
**Presentation:** 3
**Contribution:** 2
**Rating:** 5
**Confidence:** 5

**Summary:**

In this paper, the authors proposed a general model compression framework for efficient Transformer object tracking, named CompressTracker. The method adopts a novel stage partitioning strategy to divide the Transformer layers of the teacher model into different stages, enabling the student model to more effectively simulate each corresponding teacher stage. The authors also designed a unique replacement training technique, which involves randomly replacing specific stages in the student model with specific stages in the teacher model. Replacement training enhances the student model's ability to replicate the behavior of the teacher model. To further force the student model to simulate the teacher model, we combine predictive guidance and staged feature imitation to provide additional supervision during the compression process of the teacher model. The authors conducted a series of experiments to verify the effectiveness and generality of CompressTracker.

**Strengths:**

The author has clear ideas and the article is easy to understand. He proposes a general compression framework for single object tracking. This method can efficiently compress large object tracking models into small models. The author has conducted a large number of experiments to prove the effectiveness of this method.

**Weaknesses:**

The font size of the pictures in the article is too small. The author can adjust the font size appropriately to facilitate reading. The training time line in Figure 1a is blocked, resulting in incomplete display. The font size of the tables is inconsistent, for example, the font size of Tables 5, 6, 7, and 8 is too large. The abstract is redundant and can be appropriately deleted.

**Questions:**

1.The font size of the pictures in the article is too small. The author can adjust the font size appropriately to facilitate reading.

2.The training time line in Figure 1a is blocked, resulting in incomplete display.

3.The font size of the tables is inconsistent, for example, the font size of Tables 5, 6, 7, and 8 is too large.

4.The abstract is redundant and can be appropriately deleted.

5.Did the author test the speed on other devices, such as CPU?

6.Will the codebe open source?

**Limitations:**

For lightweight tracking models, the training time is too long. The author can try to find new ways to reduce the time spent on training.

---

> ### Author Rebuttal · Authors · 2024-08-06
>
> Thanks for your insightful advice. We sincerely appreciate your valuable comments and recognition of the novelty of our work. We will carefully review and modify our manisctipt based on your suggestion to improve its presentation. We greatly value your support for our work!
>
> ## Q1: Font Size in Picture
> We apologize for the inconvenience caused by the small font size. We will follow your advice and increase the font size in Figures 1, 2, 3, 4, and 5 to enhance readability.
>
> ## Q2: Blocked Figure 1a
> Thank you for pointing out our oversight! We will address and correct the issue in the revised version of the manuscript.
>
>
> ## Q3: Inconsistent Font Size in Table and Redundant Abstract
> Thank you very much for your valuable suggestions! We will reorganize the table and revise the abstract to improve the overall presentation.
>
>
>
> ## Q4: Speed on Other Devices
> We evaluated the speed of CompressTracker on an Intel(R) Xeon(R) Platinum 8268 CPU @ 2.90GHz. The results are presented in the table below. Our CompressTracker achieves the best balance between accuracy and speed.
>
> We just propose a novel model compression framework rather than a specific model. To demonstrate the effectiveness of our framework, we applied it to compress several tracking models. Due to the framework's strong generalization capabilities, other researchers can select appropriate student models based on their hardware and apply our framework accordingly. We will add a more detailed explanation of this aspect in the revised version.
>
> | Model | AUC on LaSOT | FPS(CPU) |
> |-------|-------|-------|
> | CompressTracker-2 | 60.4 | 29 |
> | CompressTracker-3 | 64.9 | 22 |
> | CompressTracker-4 | 66.1 | 18 |
> | CompressTracker-6 | 67.5 | 13 |
> | HiT-Base | 64.6 | 33 |
> | E.T.Track | 59.1 | 42 |
> | FEAR-XS | 53.5 | 26 |
>
> | Model | AUC on LaSOT | FPS(CPU) |
> |-------|-------|-------|
> | CompressTracker-M-S | 62.0 | 30 |
> | MixFormerV2-S | 60.6 | 30 |
>
> | Model | AUC on LaSOT | FPS(CPU) |
> |-------|-------|-------|
> | CompressTracker-SMAT | 62.8 | 31 |
> | SMAT | 64.6 | 33 |
>
>
> ## Q5: Code Release
> Yes! We will release our code upon acceptance! Thank you once again for recognizing and supporting our work!
>
> ## Q6: Training Time
> We confirm this slightly longer training time in Limitaion (Section 5) and we are purchasing new technique to reduce the training time and further improve the accuracy of our CompressTracker.

---

### Official Review · Reviewer_H7HB · 2024-07-22

**Soundness:** 3
**Presentation:** 4
**Contribution:** 3
**Rating:** 3
**Confidence:** 5

**Summary:**

This paper introduces CompressTracker, a general model compression framework for efficient transformer-based object tracking. CompressTracker divides the teacher model into stages corresponding to student model layers and randomly replaces student stages with teacher stages during training. It also aligns the teacher and student models using prediction guidance and feature mimicking. The framework gradually increases the probability of using student stages throughout training. CompressTracker achieves significant speed improvements while maintaining high accuracy. For example, CompressTracker-4 accelerates OSTrack by 2.17x while preserving 96% of its accuracy on LaSOT.

**Strengths:**

- Versatility: Compatible with various transformer architectures for student models.
- Efficiency: Achieves a good balance between inference speed and tracking accuracy.
- Streamlined training: Offers a single-step, end-to-end training process, simplifying the compression pipeline.

**Weaknesses:**

- Limited theoretical analysis: The paper focuses on empirical results without providing much theoretical justification for the proposed methods.
- Lack of ablation on some components: Some components of the framework are not thoroughly explored. For instance, the impact of different feature mimicking strategies is not extensively analyzed.
- Performance and Efficiency Trade-off: While CompressTracker maintains high accuracy, there's a slight performance drop compared to the original model. Training time for CompressTracker-4 (with only 4 blocks) exceeds that of the original OSTrack. This trade-off between training efficiency, inference speed, and model performance requires further optimization.
- The core idea of reducing the number of Transformer blocks is not new. Similar approaches have been used in other models like TinyViT[1] and MiniViT[2].


[1] Wu K, Zhang J, Peng H, et al. Tinyvit: Fast pretraining distillation for small vision transformers[C]//European conference on computer vision. Cham: Springer Nature Switzerland, 2022: 68-85.
[2] Zhang J, Peng H, Wu K, et al. Minivit: Compressing vision transformers with weight multiplexing[C]//Proceedings of the IEEE/CVF Conference on Computer Vision and Pattern Recognition. 2022: 12145-12154.

**Questions:**

Please refer to the weakness.

---

> ### Author Rebuttal · Authors · 2024-08-06
>
> We greatly appreciate your recognition of the efficiency and value of our work, as well as your insightful advice and comments. Other reviewers, such as Reviewer M8LK, Reviewer 8Frr, and Reviewer 2g8j, have also acknowledged the novelty and effectiveness of our approach. Our CompressTracker achieves an optimal balance between performance and efficiency. Our CompressTracker-4 obtains 66.1% AUC on LaSOT with 228 FPS.  We would be grateful if you could reconsider our work and offer your support!
>
> ## Q1: Theoretical Analysis
> Thanks for your valuable advice. We will incorporate your suggestions and add more theoretical justification in the revised manuscript. Our replacement training is similar to the dropout technique. While dropout technique reduces the over fitting by randomly dropping neurons, our replacement training strategy prevent the student model from overfitting to specific teacher stages and improve the robustness through introducing randomness. Please refer to the comment for detailed derivation. We explain the superiority of replacement training from an information-theoretic perspective. Similar to dropout, our replacement training strategy increases the training entropy by introducing uncertainty, which enhances the generalization ability of student and reduces the overfitting of the student model to specific teacher stages. Extensive experiment results further validate effectiveness of our CompressTracker. We will follow your advice and add the theoretical justification in the final version.
>
> ## Q2: Ablation Study
> Indeed, we have conducted a series of ablation experiments to investigate each component of our CompressTracker, as detailed in Section 4.4. We examine the effects of Backbone Initialization (Table 5), Decoder Initialization and Optimization (Table 6), Stage Division (Table 7), Supervision (Table 10 and Figure 3), Training Epochs (Table 8). Besides, we perform ablation studies on Replacement training (Appendix Table 11), Progressive Replacement (Appendix Table 12), and Replacement Probability (Appendix Figure 4). We also compare the training times in Appendix Table 13 and Figure 5 to highlight the training efficiency of our CompressTracker. These comprehensive experiments are designed to assess the impact of each component thoroughly.
>
> For feature mimicking strategies, we analyzed various methods including ‘L1’, ‘L2’, ‘Proj + L2’, and ‘CE’ (L1-norm, L2-norm, linear projection before L2-norm, and cross-entropy). Results are shown in the following table. he performance of different strategies varies, and the 'L2' strategy adopted by our CompressTracker does not achieve the highest accuracy. To demonstrate that the improvement comes from our model compression framework instead of the specific and complex feature mimicking strategies, we just adopt the simplest approach: L2 distance without any complicated design. Our CompressTracker achieves the best balance between inference speed and accuracy, indicating that its superiority comes from our unified framework rather than from complex loss function designs.  We believe that exploring more sophisticated strategies could further enhance performance.
>
> We will follow your suggestion and add more experiments on the components of our CompressTracker to provide a more thorough analysis in the final manuscript.
>
> | Strategy | AUC on LaSOT |
> |-------|-------|
> | L1 | 65.0 |
> | L2 | 65.2 |
> | Proj + L2 | 65.3 |
> | CE | 65.4 |
>
> ## Q3: Performance and Efficiency Trade-off
> Compared to previous works, our CompressTracker achieves the best balance between performance and efficiency. As shown in Appendix Table 13, CompressTracker-4 takes about 20 hours to train, slightly longer than OSTrack’s 17 hours, but retains 96\% of its performance on LaSOT (66.1\% AUC) while achieving a 2.17× speedup.
>
> In contrast, lightweight models like HiT, despite requiring less training time, exhibit lower accuracy. HiT matches CompressTracker-4 in speed (175 FPS) but performs worse (64.6 AUC on LaSOT) compared to CompressTracker-4 (66.1 AUC and 228 FPS).
>
> Furthermore, methods with complex multi-stage reduction, such as MixFormerV2-S, require much longer training (120 hours) but deliver inferior performance (60.6 AUC on LaSOT) compared to CompressTracker-4 (66.1 AUC), which achieves significantly better results with just 20 hours of training.
>
> Many prior works, like HiT and MixFormerV2, aim to balance accuracy and efficiency, and our framework surpasses these methods. We believe our work offers a novel perspective on this issue for the tracking field, and the trade-off is not weakness but an important area for tracking. We have confirmed the training inefficiency in the Limitation (Section 5), and we will work on to enhance accuracy and reduce training duration.
>
> ## Q4: Difference from TinyViT and MiniViT
> Our CompressTracker significantly differs from TinyViT and MiniViT. TinyViT enhances inference speed by aligning a light student outputs with teacher predictions, while MiniViT reduces model parameters through weight multiplexing without affecting inference speed. Both methods focus on image classification, whereas our framework is the first to propose a unified compression approach specifically for tracking.
>
> Our model differs from previous work in the following main ways. (1) We employs a novel stage division strategy, segmenting transformer layers into distinct stages, which TinyViT and MiniViT do not use. (2) We introduce a replacement training that randomly substitutes specific stages in student with those from teacher, unlike the isolated training of TinyViT and MiniViT. (3) Figure 3 compares different training strategies, showing our CompressTracker’s superior accuracy over methods like ‘Distill Training’ used in TinyViT and MiniViT, which highlights the advantages of our framework.
> We will include a more detailed analysis of how our approach differs from previous works and the advantages of our CompressTracker in the final version of the manuscript.

---

> ### Author Response · Authors · 2024-08-06
> **Theoretical analysis of the superiority of our replacement training**
>
> Please allow me to explain the superiority of our replacement training strategy from an information theoretic perspective. Dropout increases entropy during training by randomly dropping neurons to reduce model overfitting. Our replacement training strategy is similar to dropout technique in that it introduces the training uncertainty and increase the entropy to reduce the overfitting of the student model to specific stages of the teacher. Next, we will give theoretical derivations to show why our replacement training, similar to dropout, can increase the entropy of training.
>
> Firstly, we provide a uniform formal definition of both dropout and our replacement training. Dropout can be expressed as: if $b(p) = 1$, $h^{'} = h / p$, and if $b(p) = 0$, $h^{'} = 0$, where $h$ is the output of a specific neuron, $h^{'}$ is the output after applying dropout, and $b(p)$ is the Bernoulli sampling with probability $p$. And for our replacement training, $f_{i}$ can be defined as: if $b(p) = 1$, $f_{i} = SS_{i}(f_{i-1})$, and if $b(p) = 0$, $f_{i} = TS_{i}(f_{i-1})$, where $f_{i-1}$ and $f_{i}$ are the input and output feature of the $i$-th stage, and $SS_{i}$ and $TS_{i}$ denote the $i$-th student stage and teacher stage. Without the replacement training, $f_{i} = SS_{i}(f_{i-1})$.
>
> After providing a unified representation of dropout and our replacement training, we can give a derivation of our replacement training based on the formula of dropout. Entropy $H(X)$ can be used to measure the uncertainty of a random variable $X$, which is defined as: $H(X)= - \sum_{x} P(x)\mathrm{log}P(x),$ where $P(x)$ is the probability distribution of the random variable $X$ taking values $x$. Thus, the origin entropy of $f$ can be written as: $H(f_{i})= -\sum_{f_{i}} P(f_{i})\mathrm{log}P(f_{i}).$
>
> Because $P(f_{i}) = P(SS_{i}(f_{i-1}))$, entropy $H_{SS}(f_{i})$ of student stage is: $H_{SS}(f_{i})= -\sum_{f_{i-1}} P(SS_{i}(f_{i-1}))\mathrm{log}P(SS_{i}(f_{i-1})),$ and entropy $H_{TS}(f_{i})$ of teacher stage can be written as: $H_{TS}(f_{i})= -\sum_{f_{i-1}} P(TS_{i}(f_{i-1}))\mathrm{log}P(TS_{i}(f_{i-1})).$
>
> With our replacement training, the $P(f_{i})$ is: if $b(p) = 1$, $P(f_{i}) = pP(SS_{i}(f_{i-1}))$, and if $b(p) = 0$, $P(f_{i}) = (1-p)P(TS_{i}(f_{i-1}))$.
>
> Thus, the entropy of $P(f_{i})$ can be written as:
>
> $H(f_{i}) = -\sum_{f_{i}} P(f_{i})\mathrm{log}P(f_{i})$
>
> $= -[\sum_{f_{i-1}}(1-p)P(TS_{i}(f_{i-1})) \mathrm{log}((1-p)P(TS_{i}(f_{i-1}))) + \sum_{f_{i-1}} pP(SS_{i}(f_{i-1})) \mathrm{log}(pP(SS_{i}(f_{i-1})))]$
>
> $= -[(1-p)\mathrm{log}(1-p)\sum_{f_{i-1}}P(TS_{i}(f_{i-1}))] + (1-p)\sum_{f_{i-1}}P(TS_{i}(f_{i-1}))\mathrm{log}P(TS_{i}(f_{i-1})) + p\mathrm{log}p\sum_{f_{i-1}}P(SS_{i}(f_{i-1})] + p\sum_{f_{i-1}}P(SS_{i}(f_{i-1}))\mathrm{log}P(SS_{i}(f_{i-1}))$
>
> $= -[(1-p)\mathrm{log}(1-p)+p\mathrm{log}p] + (p-1)H_{TS}(f_{i}) + pH_{SS}(f_{i}).$
>
> Thus, the difference in entropy $\Delta H(f_{i})$ is:
>
> $\Delta H(f_{i}) = -[(1-p)\mathrm{log}(1-p)+p\mathrm{log}p]+ (p-1)H_{TS}(f_{i}) + pH_{SS}(f_{i}) - H_{SS}(f_{i})$
>
> $= -[(1-p)\mathrm{log}(1-p)+p\mathrm{log}p] + (1-p)(H_{SS}(f_{i}) - H_{TS}(f_{i})).$
>
> When $p$ is in range $[0,1]$, the term $-[(1-p)\mathrm{log}(1-p)+p\mathrm{log}p]$ is always positive. Since the student model generally performs worse than the teacher model, $H_{SS}(f_{i}) - H_{TS}(f_{i})$ is typically positive as well. Thus, $\Delta H(f_{i})$ is consistently positive. This demonstrates that our replacement training achieves a similar effect to dropout: increases training entropy.
>
> Based on the above theoretical analysis, we demonstrate that our replacement training method is similar to dropout in its ability to increase training entropy. Consequently, our replacement training helps reduce the overfitting of the student model to specific stages of the teacher.

---

### Decision · Program_Chairs · 2024-09-25

**Decision:**

Reject

**Comment:**

This paper introduces a model compression framework designed for efficient transformer-based object tracking. It received scores of 3, 5, 5, 6, and 9, with an average score of 5.6.

Reviewers recognized the paper’s strengths, particularly its introduction of a new approach for compressing transformer-based tracking models to improve efficiency (highlighted by reviewers M8LK, 8Frr, pwR6). However, they noted several weaknesses, including limited technical contributions (pointed out by reviewers H7HB, 8Frr, pwR6) and insufficient experiments (cited by reviewers H7HB, 8Frr).

The Associate Chair (AC) also reviewed the paper and identified the main contributions as a stage division strategy, which compresses the original model in a stage-wise manner, and a stage replacement training scheme, which randomly substitutes specific stages in the student model with those from the teacher model. The results indicate that while the proposed method achieves approximately a 2x speed acceleration, it incurs a performance drop of about 3% and 8% in AUC score on the LaSOT dataset using the OSTrack and MixFormer tracking models, respectively. The improvements over using a smaller backbone model are not significant, and the claim of a ‘general compression framework’ does not hold, particularly as performance drops significantly when applied to the MixFormer tracker.

After discussions among the authors, reviewers, and the AC, some concerns have been addressed. However, the issues regarding insufficient experiments (noted by reviewers pwR6 and H7HB) remain unresolved. Therefore, the AC cannot recommend accepting this paper and encourages the authors to consider the reviewers' comments for future submissions.